# A conformation-based intra-molecular initiation factor identified in the flavivirus RNA-dependent RNA polymerase

**Jiqin Wu**[1,2,3], **Han-Qing Ye**[1], **Qiu-Yan Zhang**[1,2], **Guoliang Lu**[1,2], **Bo Zhang**[1], **Peng Gong**[1,4]*

**1** Key Laboratory of Special Pathogens and Biosafety, Wuhan Institute of Virology, Center for Biosafety Mega-Science, Chinese Academy of Sciences, Wuhan, Hubei, China, **2** University of Chinese Academy of Sciences, Beijing, China, **3** Wuhan Institute of Physics and Mathematics, Chinese Academy of Sciences, Wuhan, Hubei, China, **4** Drug Discovery Center for Infectious Diseases, Nankai University, Tianjin, China

* gongpeng@wh.iov.cn

**Data Availability Statement:** All coordinates and structure factor files are available from the PDB database (accession numbers: 6KR2 and 6KR3).

**Funding:** This work was supported by the National Key Research and Development Program of China

## Abstract

The flaviviruses pose serious threats to human health. Being a natural fusion of a methyltransferase (MTase) and an RNA-dependent RNA polymerase (RdRP), NS5 is the most conserved flavivirus protein and an important antiviral target. Previously reported NS5 structures represented by those from the Japanese encephalitis virus (JEV) and Dengue virus serotype 3 (DENV3) exhibit two apparently different global conformations, defining two sets of intra-molecular MTase-RdRP interactions. However, whether these NS5 conformations are conserved in flaviviruses and their specific functions remain elusive. Here we report two forms of DENV serotype 2 (DENV2) NS5 crystal structures representing two conformational states with defined analogies to the JEV-mode and DENV3-mode conformations, respectively, demonstrating the conservation of both conformation modes and providing clues for how different conformational states may be interconnected. Data from *in vitro* polymerase assays further demonstrate that perturbing the JEV-mode but not the DENV3-mode intra-molecular interactions inhibits catalysis only at initiation, while the cell-based virological analysis suggests that both modes of interactions are important for virus proliferation. Our work highlights the role of MTase as a unique intra-molecular initiation factor specifically only through the JEV-mode conformation, providing an example of conformation-based crosstalk between naturally fused protein functional modules.

## Author summary

The function of a protein is often dictated by a single defined fold, which in turn is determined by its amino acid sequences. However, multiple global conformations can be utilized by a protein to fulfill distinct functions under different circumstances. The flavivirus NS5 protein, a natural fusion of an N-terminal methyltransferase (MTase) and a C-terminal RNA-dependent RNA polymerase (RdRP), may be such an example. Previously reported NS5 crystal structures exhibit two apparently different global conformations. In

(www.most.gov.cn, 2018YFA0507200 to PG, and 2016YFC1200400 to HQY), the National Natural Science Foundation of China (www.nsfc.gov.cn, 31670154 to PG and 31370198 to PG).The funders had no role in study design, data collection and analysis, decision to publish, or preparation of the manuscript.

**Competing interests:** The authors have declared that no competing interests exist.

this work, we demonstrate that both conformations are conserved in the flaviviruses and important for virus proliferation, but only one of them is clearly relevant to RdRP catalysis, in particular at the early stages of the RNA synthesis.

## Introduction

The flaviviruses are a large group of positive-strand RNA viruses, including dengue virus (DENV), West Nile virus (WNV), Japanese encephalitis virus (JEV), Zika virus (ZIKV), tick-borne encephalitis virus (TBEV), and Omsk hemorrhagic fever virus (OHFV). The majority of flaviviruses are mosquito-borne or tick-borne, sometimes causing human encephalitis or hemorrhagic diseases. The recent ZIKV outbreaks in South and North America, and more recently in Southeast Asia, have intensified the global threats of flaviviruses, in part due to the capabilities of the virus to cause birth defects through maternal-fetal transmission [1]. The flavivirus RNA genome is 10–11 kilo-bases in length, bearing a type 1 cap and lacking a poly-adenine tail. It encodes a large polyprotein that is further processed by viral and host proteases, yielding three structural proteins C/prM/E, and seven nonstructural proteins NS1/NS2A/NS2B/NS3/NS4A/NS4B/NS5 [2]. Being a unique natural fusion of an N-terminal methyltransferase (MTase) and a C-terminal RNA-dependent RNA polymerase (RdRP), the NS5 is the largest and most conserved protein encoded by flaviviruses. The NS5 MTase catalyzes the guanylyl-transfer and both the guanine N7 and nucleoside 2′-O methylation steps in the capping process, and is a single-domain module adopting a common S-adenosyl-L-methionine (SAM)-dependent MTase fold [3,4]. The K-D-K-E catalytic tetrad sits in the center of the MTase catalytic cleft, with the methyl donor SAM binding site and the cap binding site residing on the opposite sides. The RdRP module is the central molecular machine governing the viral genome replication, and has an encircled human right hand architecture with palm, fingers, and thumb domains surrounding the active site [5,6]. The fingers domain can be further divided into index, middle, ring, and pinky subdomains according to nomenclatures first used in describing the poliovirus (PV) RdRP (Fig 1A) [7,8]. Among the seven viral RdRP catalytic motifs, A/B/C/D/E are within the most conserved palm, and F/G are part of the ring and pinky fingers, respectively. Motifs A/B/C/F contain amino acids highly conserved in all viral RdRPs, and these conserved residues have highly analogous spatial arrangements around the polymerase active site [9,10]. Being an entity bearing two active sites and multiple essential viral enzymatic activities, NS5 has become a very attractive system in flavivirus research, and understanding the interplay between its MTase and RdRP modules is undoubtedly critical.

 Among the evidence related to MTase-RdRP crosstalk, high-resolution structures of full-length NS5 are essential in providing direct and informative readout of the interactions between the two modules. To date, two types of global conformations have been observed in full-length NS5 structures [11]. The conformation revealed by the JEV NS5 structure (named JEV-mode hereinafter) features a medium size interface (~1540 Å$^2$, for all interface area values presented in this study, the total buried solvent accessible surface from both side of the interface was accounted) with a conserved hydrophobic core [8,12], and is also observed in recently reported ZIKV, yellow fever virus (YFV), and DENV serotype 2 (DENV2) full-length NS5 crystal structures [13–18]. In such a conformation, the MTase approaches the RdRP from its backside and interacts with the RdRP middle finger, ring finger, and an index finger helix bearing part of a nuclear localization signal (NLS-helix) [19] (Fig 1A). The second conformation was observed in two different crystal forms of DENV serotype 3 (DENV3) NS5 (named DENV3-mode hereinafter) [20,21] (Fig 1B). In this case, the MTase also approaches the RdRP from the backside, but it is related to the JEV conformation by an approximately 110˚ rotation around an axis passing

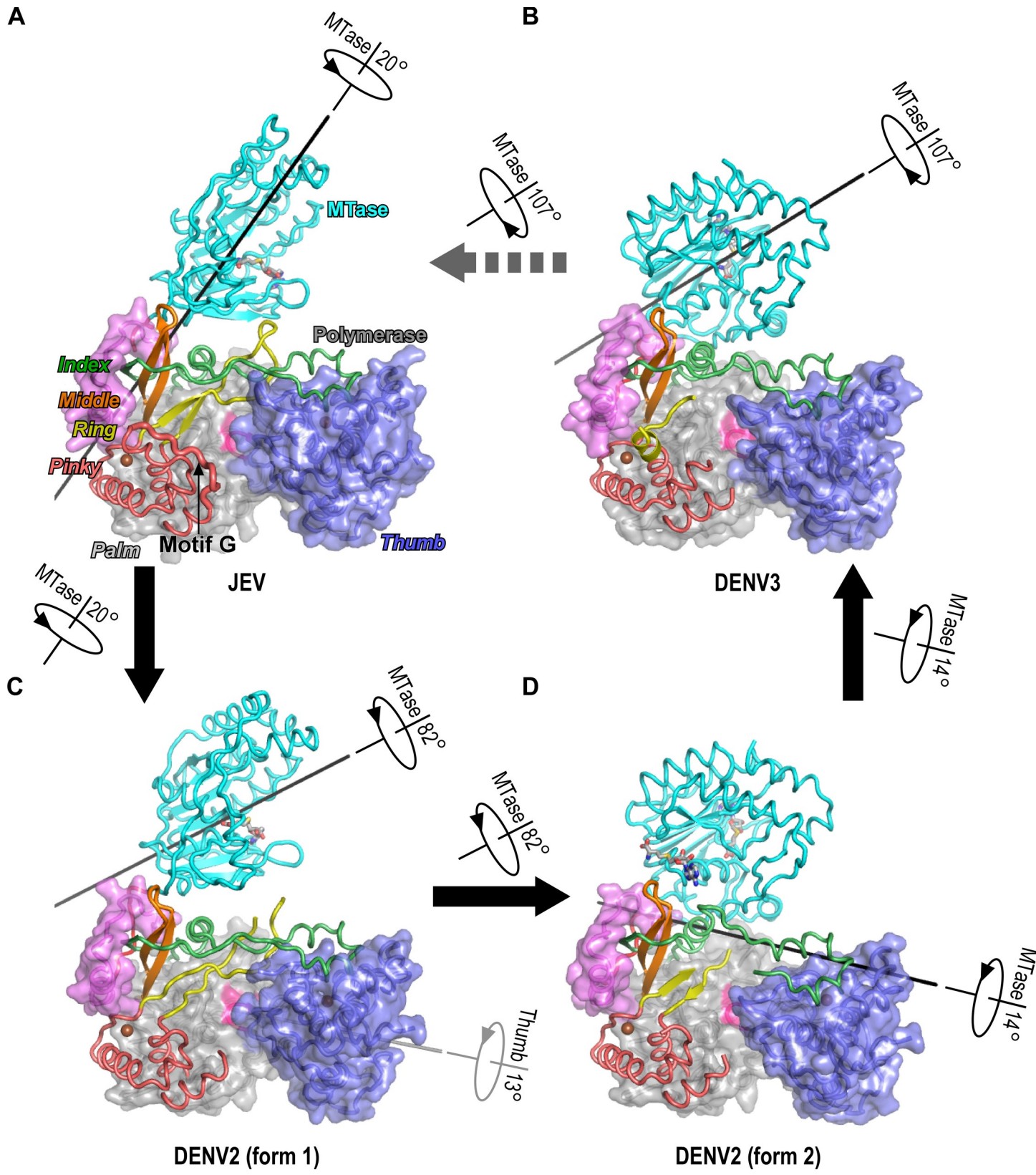

**Fig 1. Two different conformational states of DENV2 NS5 and their relationship between the JEV and DENV3 structures.** Superimposed but individually presented structures of JEV (A, PDB entry: 4K6M, chain A), DENV3 (B, PDB entry 4V0Q), and two forms of DENV2 (C and D) NS5 shown in the orientation viewing from the top

of the RdRP. Coloring scheme: MTase in cyan, linker in red, RdRP palm in grey, thumb in blue, index in green, middle in orange, pinky in light red, N-terminal extension (NE) in pink, and the signature YGDD sequence in magenta. Zinc ions and SAH molecules are shown as brown spheres and sticks, respectively. Block arrows are used to indicate plausible conformational transitions between structural states together with the straight line and rotation direction and angle associated with each transition. The fully ordered motif G region (residues 404–415) in the pinky finger of the JEV structures is highlighted by thicker ribbon representations.

near its center of mass with less than 6 Å translation. The RdRP index and middle fingers are still involved in the interactions in a slightly larger interface (~1650–1780 Å$^2$), and the nature of the interactions is instead primarily polar. Notably, the NTP binding ring finger (motif F) contacts with the MTase are absent in the DENV3 structures and the ring finger itself and the adjacent motif G residues in the pinky finger are largely disordered (Fig 1B). Motif G participates in RNA template binding and has been proposed to participate in the translocation step after every phosphoryl transfer reaction [22,23]. Hence, the JEV-mode conformation likely represents a state more suitable for polymerase synthesis from the structural perspective.

With two monomer conformation modes and eight crystal forms identified, more than 10 NS5 dimer interfaces can be recognized in the aforementioned NS5 crystal structures with no obvious conservative features. A couple of studies did focus on some of these dimer interface interactions, even though the primary NS5 solution state is monomer [16,21]. Either by probing the inter-molecular interactions, deleting the MTase domain, or mutating the MTase-RdRP linker, multiple *in vitro* polymerase assay-based studies together suggest that the MTase regulates the RdRP catalytic activities, albeit to overall moderate extents [12,14,16,18,24,25]. However, the RdRP assays established in all these studies, no matter in primer-dependent or *de novo* (including dinucleotide driven) format, did not demonstrate the formation of a processive RdRP elongation complex (EC), at least for the majority of the polymerase molecules, and all these assays require the manganese ion (Mn$^{2+}$) for catalysis, albeit in combination with the magnesium ion (Mg$^{2+}$) in some cases. In other words, the mutation-derived effect on RdRP synthesis observed in these studies may only reflect overall changes in non-processive RdRP synthesis activity, while specific alteration of either RdRP initiation or elongation cannot be clearly judged. Furthermore, none of these studies characterized both the JEV- and DENV3-mode monomer conformations to distinguish their differences in RdRP synthesis, except for our previous JEV NS5 study that only probed the JEV-mode conformation [12]. Therefore, the precise mechanism of the regulation and the explicit contribution of either NS5 conformation remain to be clarified.

In this work, we report two crystal forms of DENV2 NS5 that reveal two conformational states bearing clear analogies to those observed in the JEV-mode and DENV3-mode NS5 structures, respectively. Virological data further support the conservation and the functional importance of both conformation modes. NS5 constructs bearing mutations specifically probing two modes of MTase-RdRP intra-molecular interfaces were tested in *in vitro* polymerase assays, and only the JEV-mode interface related mutants inhibited polymerase initiation primarily through a three-fold reduction in the Michaelis constant of the initiating NTP ($K_{M, NTP}$), while polymerase EC properties were not much affected by mutations probing both modes of interactions. Collectively, our work demonstrates the conformational conservation and diversity of the flavivirus NS5 and highlights the specific contribution of the JEV-mode conformation to polymerase initiation.

## Results

### The first form of DENV2 NS5 structure contains a partially open MTase-RdRP interface and is clearly related to the JEV-mode NS5 structures

With an aim to further understand the conformational diversity of NS5 and related functional relevance, we crystallized and solved the structures of DENV2 NS5 in two different crystal

forms at 3.1 Å (form 1) and 2.9 Å (form 2) resolution (Table 1). Each structure has two NS5 molecules in the crystallographic asymmetric unit, and has the two molecules arranged in a dimer through pseudo two-fold symmetries with highly consistent global conformation (root mean square (RMS) deviation values for superimposable C-α atoms are 0.74 Å and 0.57 Å, respectively; chain A as the reference). Strikingly, the NS5 conformations between the two crystal forms are quite different (Fig 1C and 1D). Using a maximum likelihood superpositioning method [26], the RdRP palm and the majority of the fingers domain were identified as the structurally most conserved regions in the superpositioning including these and the representative full-length flavivirus NS5 structures (Fig 1). The DENV2 form 1 conformation is clearly related to the JEV-mode conformation with the JEV interface partially opened through a pure 20˚ rotation along an axis passing the vicinity of the highly conserved GTR residues that were proposed to pivot the MTase movement relative to RdRP [8,27] (Fig 1A and 1C; Fig 2A). The partial opening of the interface results in the reduction of the interface area to only about 900 Å$^2$ or 59% of the JEV interface. Among the six conserved hydrophobic residues forming the

**Table 1. X-ray diffraction data collection and structure refinement statistics.**

| Crystal form–PDB | 1 – 6KR2 | 2 –6KR3 |
|---|---|---|
| **Data collection[1]** | | |
| Space group | P2$_1$ | C222$_1$ |
| Cell dimensions | | |
| $a$, $b$, $c$ (Å) | 87.2, 146.4, 98.4 | 178.8, 210.0, 157.9 |
| α, β, γ (˚) | 90, 105.8, 90 | 90, 90, 90 |
| Resolution (Å)[2] | 60.0–3.06 (3.17–3.06) | 60.0–2.93 (3.03–2.93) |
| R$_{merge}$ | 0.160 (0.50) | 0.138 (0.55) |
| R$_{meas}$ | 0.191 (0.60) | 0.153 (0.61) |
| CC$_{1/2}$ | 0.933 (0.781) | 0.952 (0.832) |
| I / σI | 6.8 (2.1) | 11.6 (3.0) |
| Completeness (%) | 98.3 (99.1) | 99.9 (100.0) |
| Redundancy | 3.2 (3.2) | 5.6 (5.5) |
| **Refinement** | | |
| Resolution (Å) | 3.06 | 2.93 |
| No. reflections | 44,340 | 63,847 |
| R$_{work}$ / R$_{free}$[3] (%) | 22.4 / 27.5 | 23.3 / 27.0 |
| No. atoms | | |
| Protein | 12638 | 12758 |
| Ligand / Ion / Water | 52 / 4 / 33 | 134 / 19 / 47 |
| B-factors (Å$^2$) | | |
| Protein | 27.9 | 60.3 |
| Ligand / Ion / Water | 54.5 / 76.4 / 22.5 | 59.9 / 93.2 / 52.5 |
| RMS deviations | | |
| Bond lengths (Å) | 0.011 | 0.012 |
| Bond angles (˚) | 1.42 | 1.25 |
| Ramachandran stat.[4] | 79.9 / 17.2 / 1.7 / 1.2 | 83.1 / 15.1 / 1.0 / 0.8 |

[1] One crystal was used for data collection for each structure.

[2] Values in parentheses are for highest-resolution shell.

[3] 5% of data are taken for the R$_{free}$ set.

[4] Values are in percentage and are for most favored, additionally allowed, generously allowed, and disallowed regions in Ramachandran plots, respectively.

interface core in the JEV structure, only three of them (DENV2 NS5 residues W121, F349, and P583) remained as part of the interface. The tip of the ring finger no longer contacts the MTase in the DENV2 form 1 structure, and its electron density becomes weak but still readily traceable and its folding is largely consistent with the JEV conformation (Fig 1A and 1C; Fig 2A). However, the motif G region in the pinky finger is largely disordered as observed in the DENV3 structures [20,21]. Based on these observations, we propose that the interactions between the MTase residues 113 and 115 and the phenylalanine (residue 465 in DENV2 NS5) in the tip of the RdRP ring finger are essential for maintaining the canonical folding of the ring and pinky fingers, and the folding of the motif G region in the pinky finger is likely dependent on the dynamics of the adjacent ring finger. It is also worth noting that, in addition to ring and pinky fingers, the index finger is also partially disordered in most of the RdRP-only flavivirus NS5 structures [5,6]. These observations together suggest that MTase interaction likely contributes to the folding of RdRP fingers domain, which in turn could affect polymerase properties including RNA binding and subsequent catalytic events.

## The second form of DENV2 NS5 structure resembles the DENV3 NS5 structures

The DENV2 form 2 conformation is instead analogous to the DENV3-mode conformation but contains previously unidentified features. It is related to the DENV3-mode conformation by a 14° rotation along an axis near the interface and the MTase-RdRP linker region (residues 264–273 in DENV2 NS5) and a translation less than 2 Å (Fig 1B and 1D; Fig 2B). The primarily polar interactions between the NLS-helix and the MTase are largely retained (Fig 2B), and interface area is about 1460 $Å^2$ and is only reduced for about 11% as compared to the first reported DENV3-mode interface [20]. The rotational movement widens the cleft between the relatively conserved residue pair E67-R68 in the MTase and the RdRP middle finger, creating a pocket that allows the binding of a putative SAH molecule with high occupancy (0.90 and 0.97) in addition to the SAH molecule usually bound in the SAM binding pocket of the MTase (Fig 1D; Fig 2B and 2C). Usually, SAH can be co-crystallized with the flavivirus MTase at a 1:1 molar ratio with the SAH co-purified with the MTase after its overexpression in bacterial culture [4,8,20,28]. It is possible that the observed NS5 conformation allowed SAH co-purification with NS5 at a higher molar ratio. Such a secondary SAH binding site has not been observed in numerous MTase-containing flavivirus NS5 structures. This binding pocket appears to be not tight enough, as the SAH molecules bind at moderately different positions in the two NS5 proteins in the crystallographic asymmetric unit, and specific interactions between the non-carbon atoms of the SAH and the side chains of the NS5 are largely lacking (Fig 2B and 2C). Nevertheless, based on the fact that this secondary SAH binding pocket is created at the MTase-RdRP interface and is reasonably conservative (Fig 2C), it might have potential as a target to develop small molecule inhibitors against flaviviruses.

## The correlation among all four NS5 conformational states and the structural elements that may mediate NS5 conformational dynamics

The two DENV2 NS5 structures nicely fill the gap of major conformational differences between the JEV-mode and DENV3-mode structures, suggesting a plausible order from JEV-mode to DENV2 form 1, then to form 2, and finally to DENV3-mode, primarily through rotational movements in more or less consistent directions (Fig 1; Fig 3A and 3B; S1 Movie). Three structural elements may play critical roles in the transitioning among these states. The first is the universally conserved GTR sequence at the C-terminal end of the MTase (Fig 3C). This tripeptide sequence was proposed as a pivoting element in the work of the full-length JEV

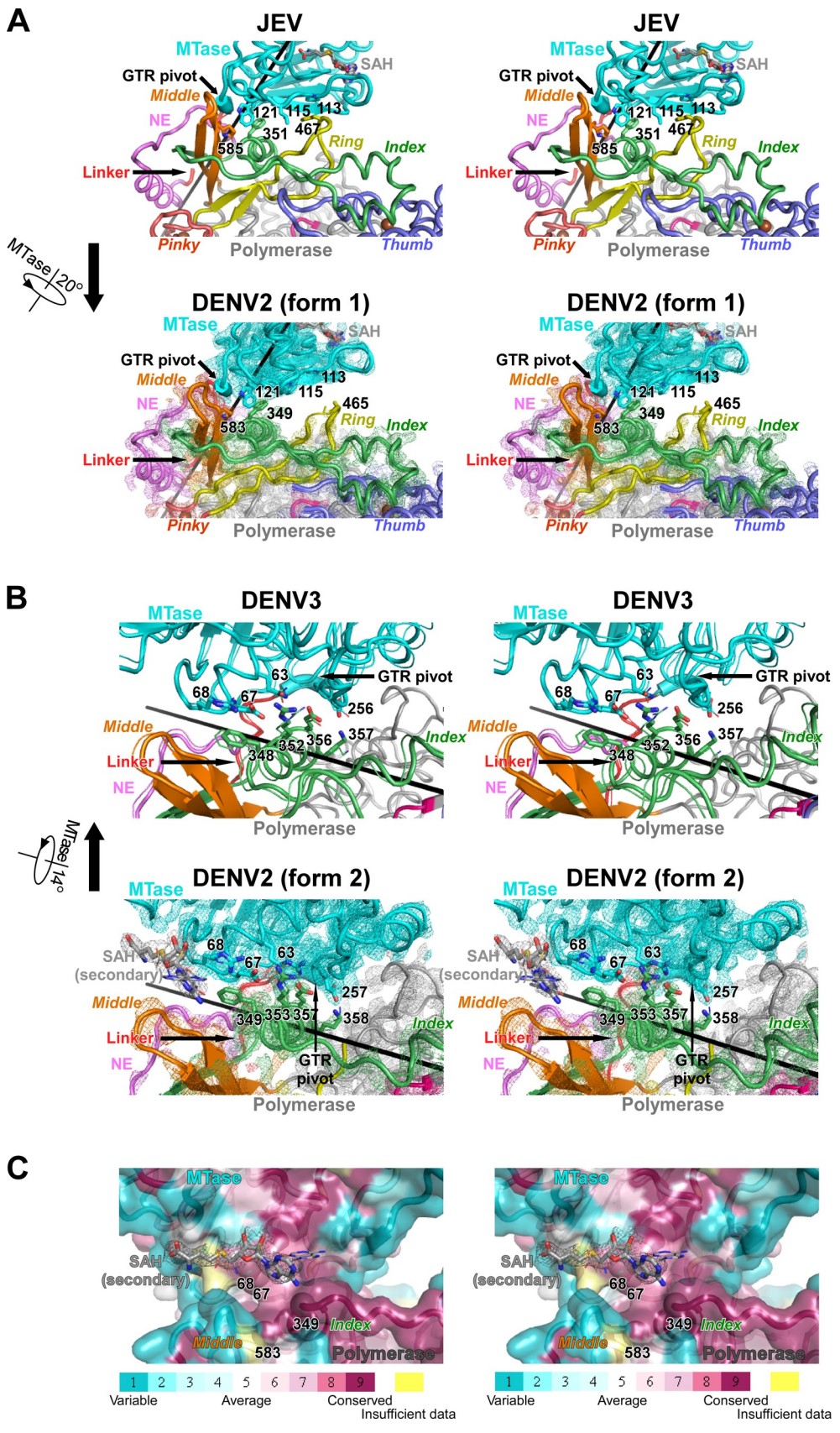

**Fig 2. A comprehensive comparison of the intra-molecular MTase-RdRP interface shown as stereo-pair images.**
A) A comparison between the JEV (top) and the first form of DENV2 (bottom) structures. B) A comparison between the DENV3 (top, two models) and the second form of DENV2 (bottom) structures. C) The binding of the second SAH molecule observed in the form 2 of DENV2 structure. The binding pocket is shown as surface representations with conservation scores projected. Thinner sticks show the moderately different SAH binding mode observed in the other NS5 molecule in the crystallographic asymmetric unit. Composite simulated-annealing (SA) omit electron density maps contoured at 1.2 σ are overlaid with the DENV2 models in panels A and B and the SAH molecule in panel C. For the DENV2 structures in panels A and B, the two NS5 molecules in the crystallographic asymmetric unit were superposed and shown as thick and thin representations with the density maps of the thick model overlaid. All structures in panels A and B were superposed but may be presented separately. The coloring scheme is the same as in Fig 1. For panels A and B, the rotational movements correlate the both structure pairs are indicated.

NS5 structure and was proved to be functionally important in both JEV and DENV2 replications [8,27]. The second element is the 10-residue MTase-RdRP linker that overall exhibits low sequence conservation in flaviviruses (Fig 3C). A comparison of all full-length NS5 structures demonstrated that the N-terminal half of the linker undergoes a swinging motion with a partial refolding to become helical when transitioning from the JEV-mode conformations to the DENV3-mode ones, while the C-terminal half remains unaffected (Fig 3A and 3B). Not surprisingly, mutations in the linker region or linker substitutions using sequences of other flavivirus NS5 have been found to affect NS5 conformation distribution, NS5 enzymatic activities, and virus proliferation [18,24,29], possibly by altering the flexibility of the linker. The third element is the NLS-helix (residues 348–358 in DENV2 NS5) in the RdRP index finger. On one hand, this helix is critical to both the JEV-mode and DENV3-mode of interface interactions by contributing the F349 and R353 residues (Fig 2A and 2B; Fig 3B). On the other hand, it is at the central region of a long stretch of conserved sequences (residues 341–366 in DENV2 NS5) that may also be related to NS5 nuclear localization, nuclear export, and interaction with another important viral protein NS3 [19,30,31], emphasizing its possible importance when not participating in the intra-molecular MTase-RdRP interactions. We propose that the largely rotational movement of the MTase from the JEV-mode conformations to the DENV3-mode conformations may utilize this highly conserved helix as a guiding track (S1 Movie). At the starting and end points of the movement, the conserved hydrophobic residue patch P113/X115/W121 (the majority of X are L and M) and the conserved polar residue pair E67-R68 provide the anchoring points on the MTase side (Fig 3B–3D). On the other hand, both of these stable conformations make the NLS-helix inaccessible to other factors, and the helix may only become solvent exposed at certain stages of virus life cycle.

## Virological data support the functional relevance of both the JEV-mode and the DENV3-mode conformations

Previously, we tested the functional significance of the JEV-mode conformation using both JEV and DENV2 systems [27]. When arginine/aspartic acid/serine (R/D/S) mutations were introduced at the six hallmark hydrophobic residue sites, virus proliferation was significantly inhibited (the corresponding mutation sites in DENV NS5 were listed in Fig 3E). In order to understand functional relevance of the DENV3-mode conformation, we designed five mutations at the NS5 residues 67 and 68 for each virus system (E67A, E67D, K68A, K68R and E67A/K68A in JEV; E67A, E67D, R68A, R68K and E67A/R68A in DENV2, and correspond to the same mutations with an "M_" prefix in Fig 3E) and compared the mutant constructs with the wild type (WT) viruses (Fig 4). The residues 67–68 were chosen as the mutation sites because these two residues are highly conserved among the MTase residues that participate in the DENV3-mode MTase-RdRP interface but are not involved in the JEV-mode interface interactions. We first introduced each mutation into a JEV infectious clone [32]. After viral

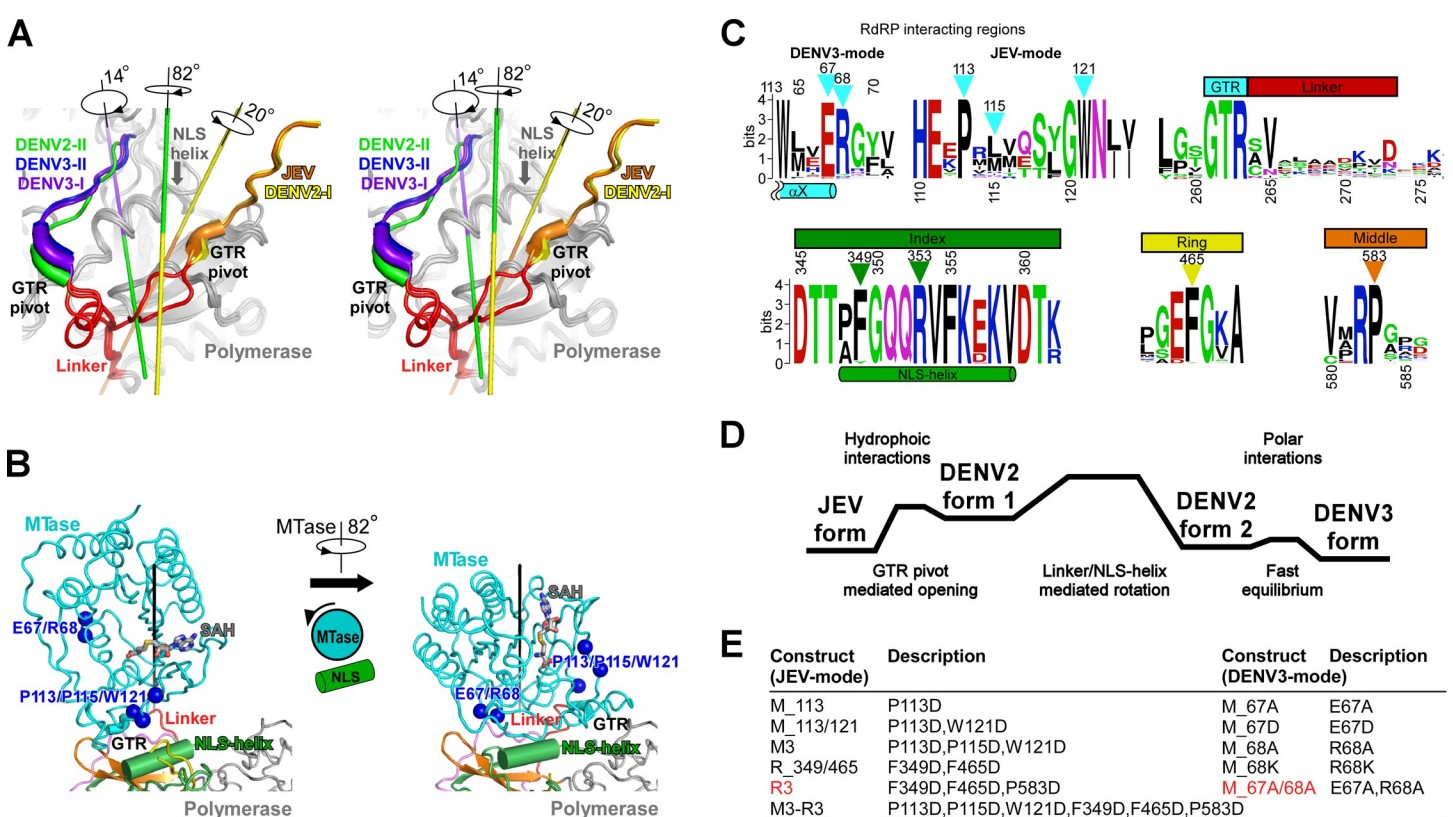

**Fig 3. Key elements that may mediate the conformational switches of the flavivirus NS5.** A) Stereo-pair images (wall-eyed) shown in a view looking at the MTase-RdRP linker region. For clarity, the entire RdRP region is shown in grey, linker is in red, and only the C-terminal ten residues including the very C-terminal GTR sequence of the MTase are shown. B) A comparison between the two DENV2 conformational states highlights a likely role of the highly conserved index finger NLS-helix. Left: JEV-mode; right: DENV3-mode. The coloring scheme is the same as in Fig 1. The α-carbon atoms of two important patches of residues are shown as blue spheres. C) The sequence logo plot showing the conservation of the two RdRP interacting regions, the GTR-linker region (top panel), and the NLS-helix that is important for both JEV- and DENV3-mode conformational states, the middle and ring finger regions only critical in JEV-mode states (bottom panel). The triangles indicated key residues involved in the intra-molecular interactions. D) A schematic free energy diagram for all four conformational states of flavivirus NS5. The relative free energy was crudely estimated by the solvent accessible surface area occluded by the NS5 intra-molecular interface interactions. E) A list of DENV2 NS5 mutants that were designed based on both the JEV- and DENV3-mode conformations, with abbreviations and full descriptions, including mutation site and mutation type.

RNA was transfected into baby hamster kidney cells BHK-21, viral protein expression and virus production were monitored. The expression level of the viral envelope (E) protein in transfected cells was detected by an immunofluorescence assay (IFA) (Fig 4A). Both the E67A and K68R mutant viruses produced similar IFA positive cells in comparison with the WT virus (100% IFA positive cells observed at 72 h post transfection (hpt)); The K68A and E67A/K68A mutants showed only around 10% positive cells; the E67D mutant produced very few IFA-positive cells. Virus productions were then quantified by a plaque assay at three time points (48, 72, and 96 h) post transfection. Consistent with the IFA data, the E67A and K68R mutant RNAs yielded similar amounts of viruses as the WT at each time point, the K68A and E67A/K68A mutants moderately impaired virus production, and viruses derived from the E67D mutant RNA-transfected cells were only detected at 72 and 96 hpt (Fig 4A). Overall, the results indicated residues 67 and 68 are important for JEV proliferation. We also performed similar analyses using a DENV2 infectious clone (Fig 4B) [33]. No IFA-positive cells were observed in the R68A, R68K and E67A/R68A transfected cells; the E67A and E67D produced around 70%-80% IFA-positive cells relative to the WT. Data from the plaque assay indicated that virus production was blocked by the R68A, R68K, and E67A/R68A mutations, while the E67A and E67D mutations had slightly less effect on virus production at each time point post

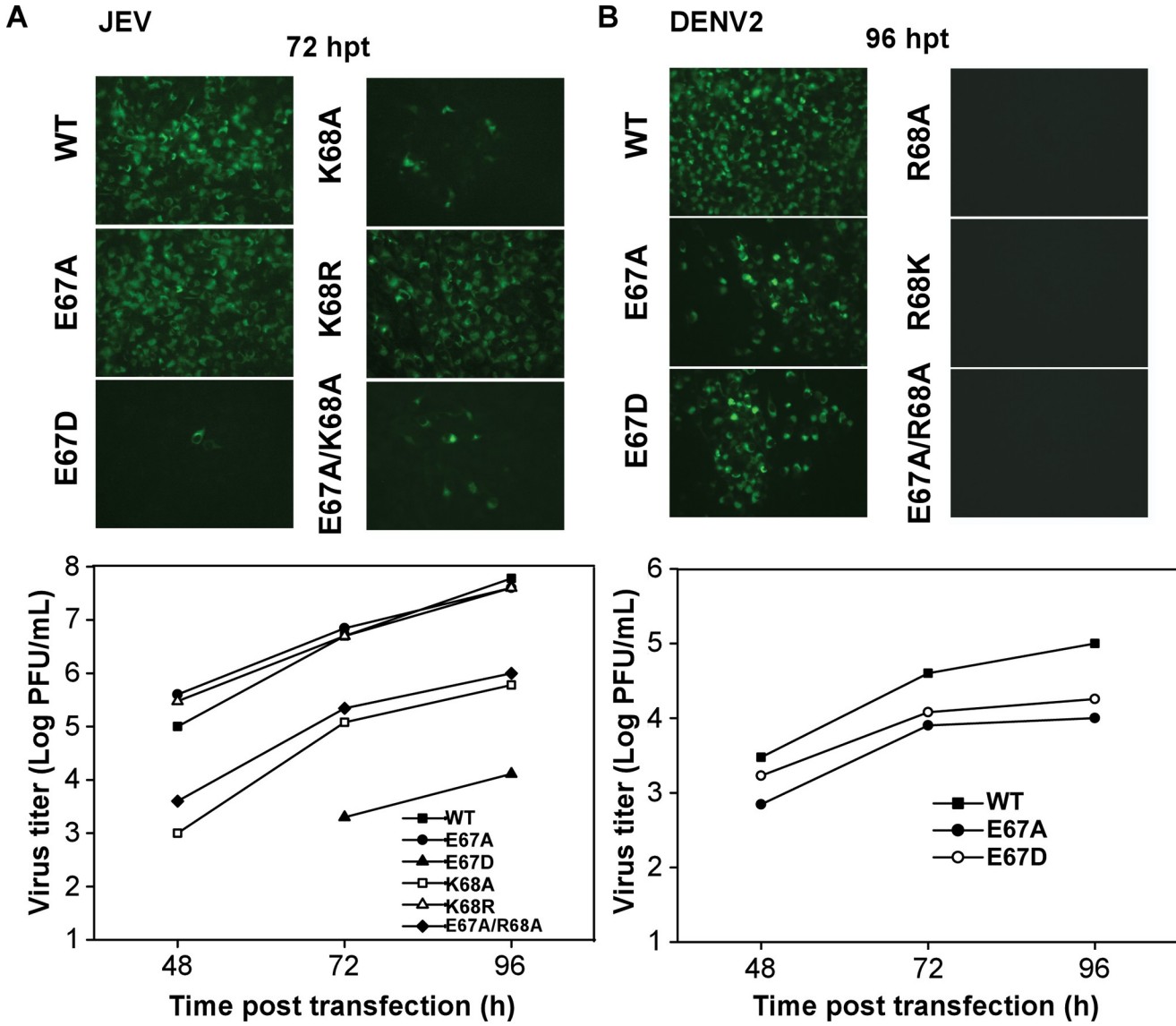

**Fig 4. Virus proliferation analyses of the WT JEV and DENV2 and the viruses bearing the DENV3-mode mutations.** A) Effects of NS5 mutations on JEV replication. IFA of JEV genome-length viral RNA containing E67A, E67D, K68A, K68R, and E67A/K68A mutations in transfected BHK-21 cells at 72 hpt. Monoclonal antibody 4G2 against envelope protein and FITC-conjugated goat anti-mouse IgG were used as primary and secondary antibodies for IFA, respectively. B) Effects of NS5 mutations on DENV2 replication. IFA of DENV2 genome-length viral RNA containing different mutations of NS5 in transfected BHK-21 cells at 96 hpt. Virus production of the supernatants of the transfected cells at each time point post transfection was detected by monolayer plaque assay, and the visible plaques were used to calculate titers.

transfection comparing with the WT (Fig 4B). Taken together, these virological data suggest that the DENV3-mode conformation is also functionally important and is likely conserved in flaviviruses, consistent with our structural observation of both conformations in DENV2 NS5.

### The establishment of effective *in vitro* RdRP assays to characterize both initiation and elongation in DENV2 NS5

We previously tested the JEV-mode interface mutants in JEV NS5 using *in vitro* assays derived from an HCV study [12,34]. While the HCV assays have $Mg^{2+}$ as the only divalent metal ion

and allow the formation of processive ECs [34], the JEV assays require $Mn^{2+}$ for RdRP activity and the stability and reactivity of the assembled complexes are far from optimal [8]. When the HCV assay format was used in the DENV2 NS5, the RdRP enzyme behavior is consistent with the HCV enzyme, and therefore the DENV2 assays established in this study can serve as effective systems to assess RdRP catalytic properties and to analyze whether the interface interactions observed in both the JEV-mode and the DENV3-mode conformations modulate RdRP catalysis. Note that dinucleotide-driven assays, such as those in the HCV study [34], have been used in multiple polymerase systems to reasonably mimic the *de novo* initiation process [35,36]. When ATP and UTP were provided as the only NTP substrates, a GG dinucleotide primer (P2) was extended to yield a 9-mer product (P9) as directed by a 30-mer template (T30) after a 45-min incubation (Fig 5A). We assessed the reactivity of the P9-containing complex by a single-nucleotide extension assay. Because the complex was in the precipitate form under the low-salt reaction condition (20 mM NaCl) but was soluble under high-salt condition (e.g. 190 mM NaCl), we removed the excess ATP and UTP by centrifugation, pellet wash, and pellet resuspension, and then added CTP to allow the single-nucleotide addition to make a 10-mer product (P10). It turned out that the conversion from P9 to P10 was very rapid and was completed immediately after manual mixing on ice without further incubation ("0 min"; Fig 5A, lane 3). Although not accurately determined, the expected catalytic rate constant of the P9-containing polymerase complex is at least magnitudes larger than that determined in the JEV study (0.14 min$^{-1}$ for WT NS5) [12]. This observation strongly suggests that the P9-containing complex of DENV2 NS5 has completed the transition from initiation to elongation and is a *bona fide* EC. Therefore, the production of this P9-containing EC (EC9) can be used to assess the overall process of initiation followed by the transition to elongation.

## Perturbing the JEV-mode but not the DENV3-mode intra-molecular interactions impairs the DENV2 NS5 RdRP initiation but not elongation

To test the impact on polymerase catalysis brought by the intra-molecular MTase-RdRP interactions, we made two sets of DENV2 NS5 mutants. The first set contains equivalent mutations utilized in the JEV study to perturb the hydrophobic JEV-mode interface [12], and the second set that contains mutations at residues 67–68 was used to probe the polar DENV3-mode interface (Fig 3E). We compared the EC9 formation of these mutants with the WT NS5 at three incubation time points. For the WT NS5, only limited 9-mer accumulation was observed beyond the first time point (15 min), and small amount of misincorporation-related 10-mer products became obvious at the last time point (90 min) (Fig 5B, lanes 11, 18, and 25; Fig 5C, lanes 41, 47, and 53). These observations again suggest that: although the EC9 formation is a slow process due to slow pre-initiation and initiation steps, it likely produces a stable EC that is not turning over to carry out multiple rounds of P9 synthesis. Among the six JEV-mode NS5 mutants, three of them exhibited obvious slower accumulation of P9 products, in particular at shorter incubation time points (Fig 5B, compare lanes 15–17, 22–24, 29–31 to the WT lanes). In contrast, all five DENV3-mode NS5 mutants showed very similar trend of P9 accumulation as the WT enzyme (Fig 5C). These data together suggest that perturbing the JEV-mode but not the DENV3-mode interactions inhibits the overall process to produce a processive EC.

To further dissect the mechanism of inhibition brought by the JEV-mode interface mutations, we compared representative NS5 mutants (R3 for JEV-mode interface and M_67A/68A for DENV3-mode interface) with the WT enzyme in a P2-driven initiation assay to explicitly assess the continuous production of the 3-mer (P3) when ATP was provided as the only NTP substrate, while the P9 synthesis was monitored in parallel for comparison (Fig 5D). Here we

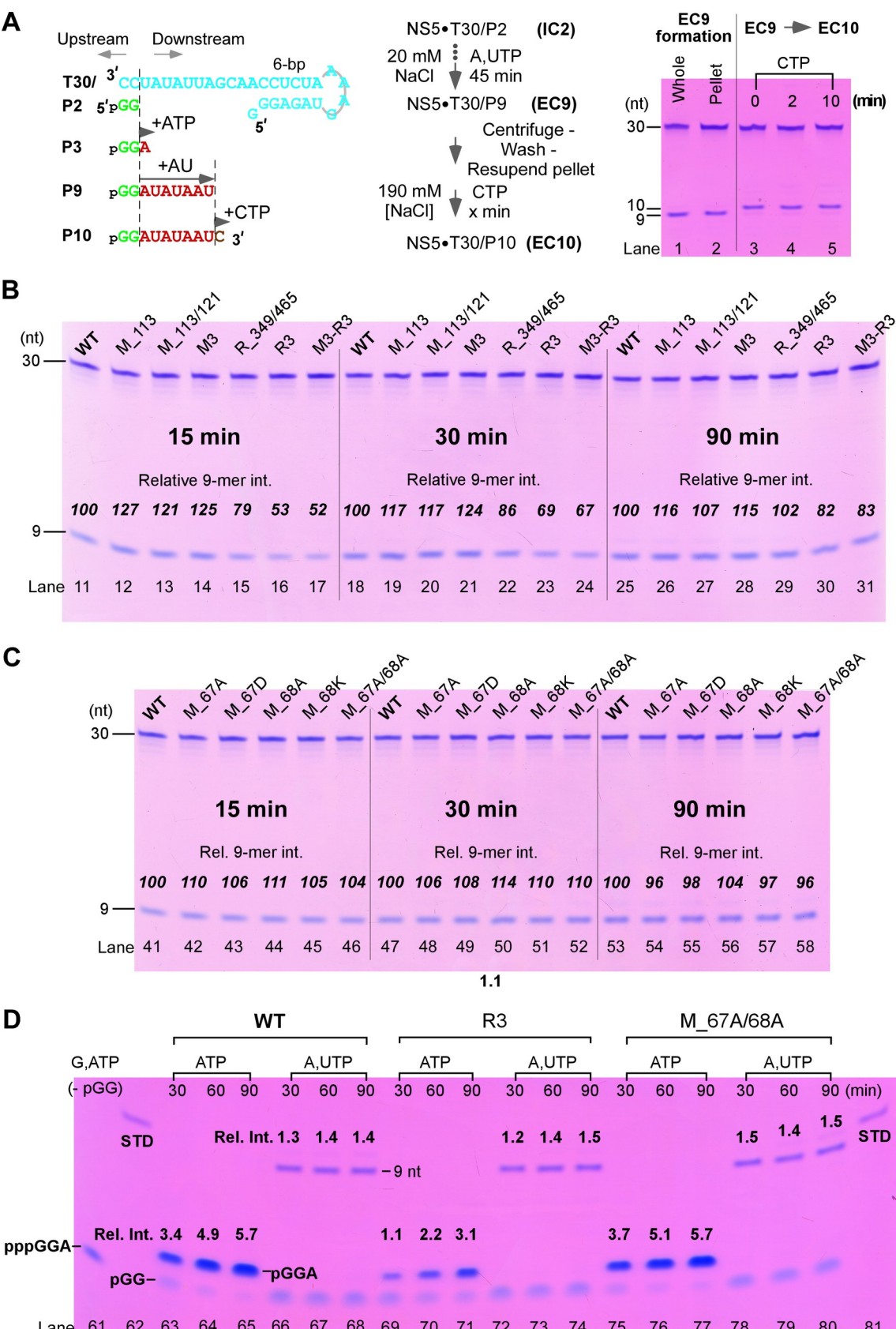

**Fig 5. Mutations perturbing the JEV-mode interface interactions impaired DENV2 RdRP initiation.** A) Left: A diagram of construct T30/P2 used in all polymerase assays and related NTP-driven reactions to generate products with different lengths. Middle: reaction flow chart of the P2-driven EC formation (IC2 to EC9) and the subsequent single-nucleotide extension (EC9 to EC10). Right: the EC9 was in a form of precipitate and was able to extend to EC10 upon CTP addition under high-salt condition. B-C) The EC9 formation comparison with the WT NS5 for the JEV-mode (B) and DENV3-mode (C) mutants. The relative intensity of the 9-mer was used to estimate the polymerase activities (the WT value for each time point series was set to 100). D) Comparison of the WT and two representative mutants (R3 for the JEV-mode; M_67A/68A for the DENV3-mode) in the multiple-turnover P3 formation and in the single-turnover P9 formation assays. The pppGGA was synthesized when GTP and ATP were provided as the only NTP substrates using the P2-free T30 template and was used as a migration marker. Note that pppGGA migrated at similar position as pGG, but faster than pGGA since it contains two extra phosphate groups at the 5′ end. A chemically synthesized 9-mer loaded with an equal molar amount to T30 was used as a quantitation standard (STD, lanes 62 and 81). The average intensity of the STD bands was set to 1. Based on previously reported evaluation, the intensity-molar amount starts to deviate from a linear relationship when the relative intensity approaches 4–5 under similar experimental settings [12]. Therefore, the intensity reported in lanes 64–65 and 76–77 are underestimated. The P9 product migrated faster than the chemically synthesized STD RNA due to its 5′-phosphate inherited from the pGG dinucleotide.

use a chemically synthesized 9-mer as a quantitation standard (STD; Fig 5D, lanes 62 and 81) loaded with an equal molar amount of the T30 template. For the WT, the R3 mutant, and the M_67A/68A mutant, the P9 amount was relatively consistent at the 60 and 90 min time points (Fig 5D, compare lanes 67, 73, 79 to lanes 68, 74, 80), suggesting that all three constructs had formed stable ECs and did not turn over to accumulate the P9 products over time. In contrast, the P3 accumulation proceeded continuously during the same period for all three constructs, indicating that the P3-containing complex is an initiation complex (IC) that carried out multiple rounds of synthesis in an abortive fashion (Fig 5D, lanes 63–65, 69–71, and 75–77). The R3 mutant had a slower P3 accumulation than the WT and the M_67A/68A mutant had, clearly suggesting that the perturbation of the JEV-mode interface impaired the RdRP initiation process.

We next performed two tests regarding the EC9 properties using the WT, R3, and M_67A/ 68A constructs (Fig 6). To test the EC9 reactivity, we compared the P9-to-P10 conversion under high and low CTP substrate concentrations for these constructs (Fig 6B). At 300 μM CTP concentration, all three constructs converted the majority of P9 to P10 at "0-min" time point (89–94% converted suggested by intensity-based quantitation) (Fig 6B, lanes 12, 14, and 16). When CTP was supplied at 5 μM, the conversion became slower, but all three constructs showed consistent progress of conversion (67–70% and 84–88% converted at "0-min" and at 1 min, respectively) (Fig 6B). To test the stability of the EC9, we used NaCl as the challenging agent in a high-salt challenge stability assay similar to those described in previous work characterizing the PV, HCV and the classical swine fever virus (CSFV) RdRPs [34,37,38]. We found that EC9 formed by all three constructs were quite stable and exhibited comparable inactivation rate constants (0.02–0.04 h$^{-1}$, corresponding to half life values of 18–35 h) upon a NaCl challenge at 500 mM concentration (Fig 6C and 6D). These data together suggest that the EC9 is highly stable and reactive, and these properties were not much affected by both types of mutations.

## The representative R3 mutant perturbing the JEV-mode interface leads to about 4-fold reduction in initiation efficiency and primarily by affecting the initiating NTP binding

To specifically investigate the enzymatic properties of the WT and the two representative mutants during the conversion of the P2 to P3, we measured the relative catalytic rates under different ATP concentrations for each construct (Fig 7A–7C), and these data were used to determine the Michaelis constants ($K_M$) of these constructs (Fig 7D–7F). By optimizing the reaction time points selection for each NTP concentration and each construct, the 3-mer band intensities were controlled to be within the linear range of the Stains-All based staining method, to facilitate quantitation accuracy (refer to our previous analysis in the JEV study

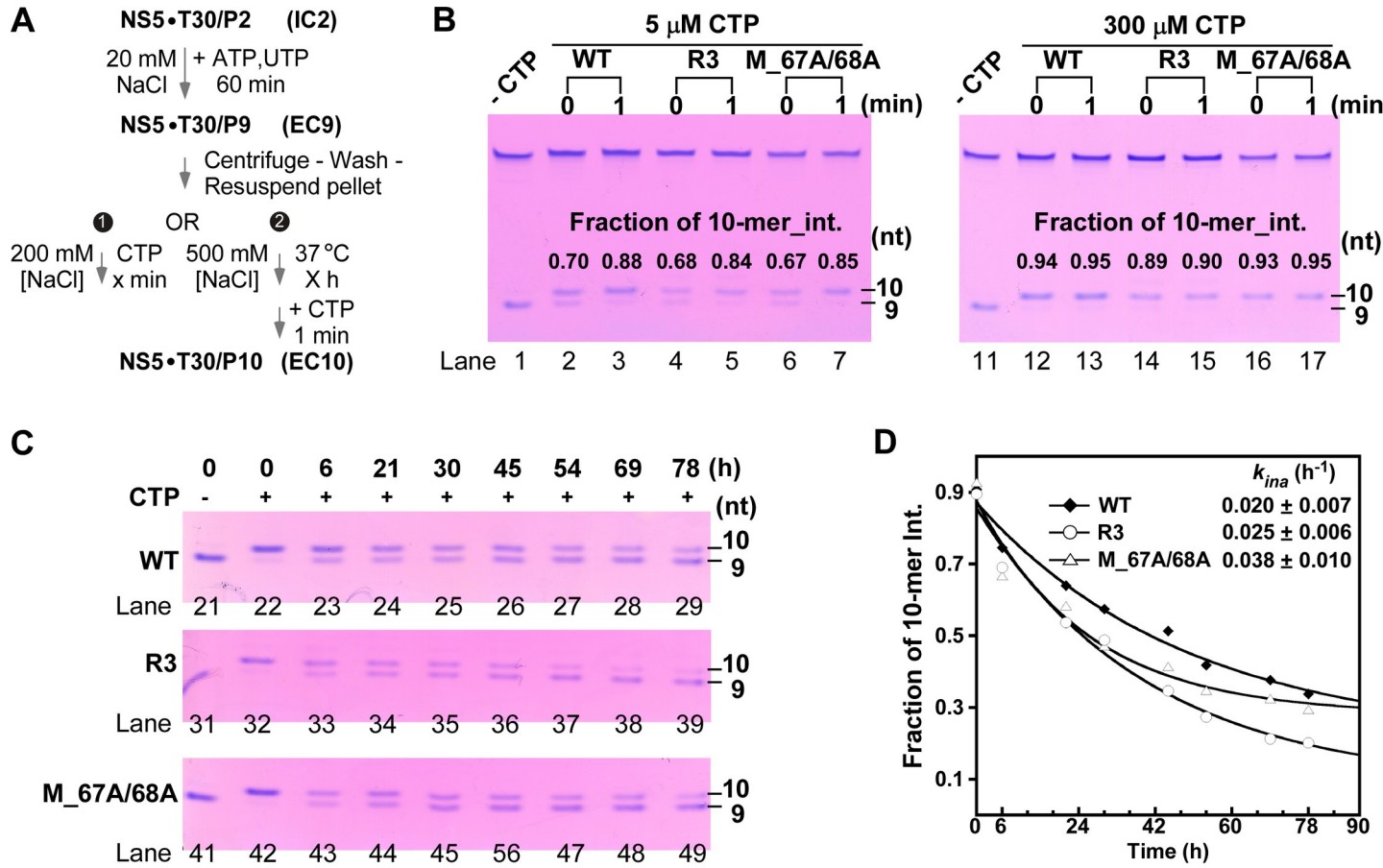

**Fig 6. The reactivity and stability of DENV2 NS5 EC9 were not apparently affected by both types of mutations.** A) The reaction flow chart for the reactivity test (1) and the stability test (2). (B) A comparison of the EC9 to EC10 conversion for the WT and two representative mutants at 5 or 300 μM CTP concentration. The fraction of the 10-mer intensity was shown in each lane. C-D) A comparison of the EC9 stability upon high-salt challenge for the above three NS5 constructs. The fraction of 10-mer intensity (determined based on gels in panel C) as a function of challenge time was plotted (D) to estimate the apparent EC inactivation rate constant ($k_{ina}$) for each construct.

[12]) (Fig 7A–7C). It turned out that the $K_M$ value of the R3 mutant is about three times the values of the WT and M_67A/68A mutant (1046 μM vs. 344 μM and 312 μM), indicating that the initiating NTP binding is clearly impaired by the JEV-mode interface mutations (Fig 7D–7F). Higher NTP concentrations were not tested in these trials due to a substrate inhibition effect observed in preparatory experiments. We then use the relative catalytic rates determined at 1000 μM for the WT and M_67A/68A mutant, and 1500 μM for the R3 mutant, respectively, to correlate the curve fittings of all three constructs (Fig 7G, see Materials and Methods). The estimated relative specificity constant (rel. $k_{cat}/K_M$) were approximately 0.27 and 1.45 for R3 and M_67A/68A mutants, respectively (Fig 7G, using the WT as reference). Taken together, these data suggest that perturbation of the JEV-mode interface clearly impaired DENV2 RdRP initiation mainly by affecting the initiation NTP binding, while perturbation of the DENV3-mode interface only had a moderate effect.

In the multiple-turnover P3 accumulation process, either the catalysis of the P2-to-P3 conversion and the dissociation of the P3 product could be rate limiting. Hence, the observed $k_{cat}$ related differences in the P3 conversion initiation assay could be affected by the variation in the P3 dissociation rates among different constructs. To further validate our judgment, we monitored the P2-driven P9 formation for the WT, R3, and M_67A/68A constructs (Fig 8). In

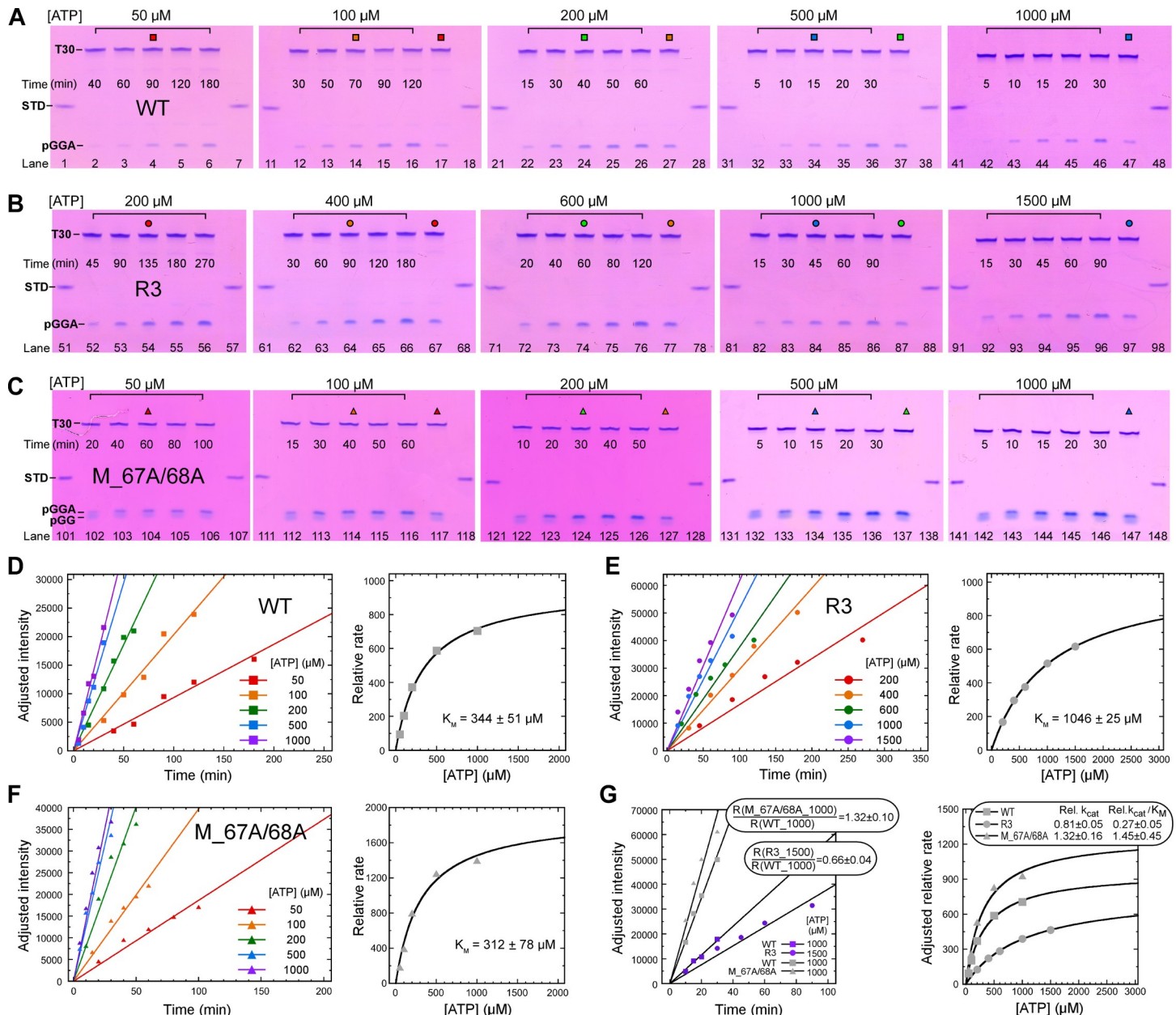

**Fig 7. Characterization of the initiation kinetics on the WT DENV2 NS5 and representative NS5 mutants.** A-C) The P3 (pGGA) formation under different ATP concentrations by the WT (A), R3 (B), and M_67A/68A (C) constructs. The color-coded icons above the lanes indicate samples from the same reaction mixture that were used to correlate intensity in different gels (see Materials and Methods). The STD samples were used as a reference and NTP concentration range was adjusted for each NS5 construct, both for ensuring that all P3 band intensities were within the linear range of the staining method used. D-F) The relative reaction rates (left) and $K_M$ fitting (right) analyses for the WT (D), R3 (E), and M_67A/68A (F) constructs. Left: The adjusted intensity of the 3-mer products as a function of time for WT, R3, and M_67A/68A with five ATP concentrations. (G) An analysis of the relative specificity constants of the R3 and M_67A/68A mutants to the WT. Single-gel based reaction rate correlation analysis (left) for the R3-WT (purple points) and M_67A/68A-WT pairs to correlate three Michaelis-Menten curves for determination of the relative specificity constants (right).

this case, the observed difference in the P9 formation is highly dependent on the catalytic rate (presumably the P2-to-P3 conversion step), while not much related by dissociation rates at subsequent steps due to the single-turnover feature of the reaction and the absence of the intermediate products (3-8-mer) in the denaturing gel analyses. When the initiating ATP was

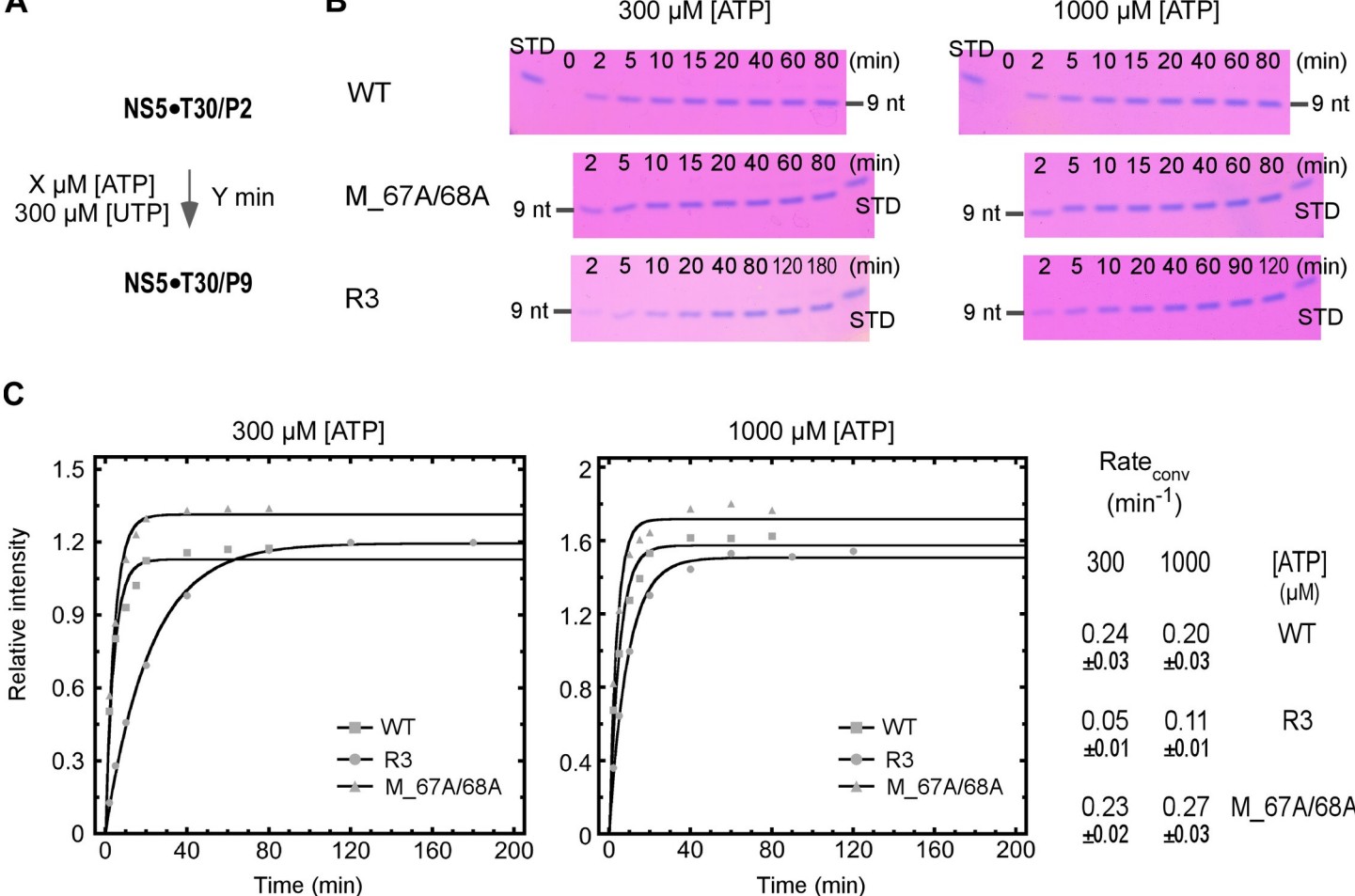

**Fig 8. A comparison of P2 to P9 conversion of the WT DENV2 NS5 constructs confirmed the initiation impairment in the R3 mutant.** A) Reaction flow chart. Two ATP concentrations (300 and 1000 μM) were tested. B) The P9 accumulation was monitored over time for the WT, R3, and M_67A/68A constructs and the STD samples were used for quantitation (set to 1). C) The relative P9 intensity as a function of time was plotted for all three constructs under two ATP concentrations. The overall conversion rate (rate$_{conv}$) was estimated by fitting each data set to a single exponential rise model.

supplied at the $K_M$-level concentration of the WT and M_67A/68A mutant (300 μM), the overall conversion rate (P2-to-P9) of the R3 mutant was much slower than those of the other two constructs (0.05 min$^{-1}$ vs. 0.23–0.24 min$^{-1}$). When the ATP concentration was lifted to the $K_M$-level of the R3 mutant (1000 μM), the conversion rate of the R3 was still lower than those of the other two constructs, while the difference between the R3 and WT/ M_67A/68A constructs became smaller (0.11 min$^{-1}$ vs. 0.20–0.27 min$^{-1}$). These data validate the judgment derived from the P3 formation assay and strongly suggest that the JEV-mode interface interactions are critical in NS5 initiation, and in particular, in the initiation NTP binding.

## Discussion

### On the conformation diversity and conformation-function relationship of the flavivirus NS5

By solving the DENV2 NS5 crystal structures in two different global conformations and characterizing the functional relevance of both conformations, the work presented here helps the

understandings of the conformational diversity and conservation in flavivirus NS5, highlighting the important role of the MTase module, a natural fusion partner of the NS5 RdRP module, in the initiation phase of RNA synthesis. Interestingly, the MTase only facilitates RdRP initiation through the JEV-mode conformation, and this specific mechanism is in agreement with the structural observation that the MTase stabilizes the folding of the NTP-binding ring finger only through this conformation [8,39] (Fig 1A). Both conformation modes seem not to apparently contribute to the RdRP catalysis in the elongation phase, supporting our previous proposal that the RdRP may only need the assistance from the MTase in the unstable initiation phase [12,40]. Based on the virological data, both conformation modes are important for virus proliferation. Therefore, the DENV3-mode conformation could contribute to other NS5-involved processes. However, whether and how it is related to the catalysis of the MTase, or to the interactions with other viral proteins or host factors remain to be clarified. Note that, the index finger NLS-helix (residues 348–359 in DENV2 NS5), possibly participating in NS3-binding or NS5 shuttling between cytoplasm and nucleus, was occluded by the MTase in both JEV- and DENV3-mode conformations [19,30,41]. Interestingly, residues in the same helix were found to mediate RANTES (regulated on activation, normal T cell expressed and secreted, also known as CCL5) expression in TBEV [31]. Therefore, additional functional relevant conformation modes likely exist, as also suggested by the small-angle X-ray scattering and reverse genetics data [5,42,43]. Based on the conformational variation of the N-terminal half of the linker region observed in crystallography, the MTase is able to reach the fingers side without much difficulty. However, a conformational change of the entire linker, or additional rearrangements of the N-terminal extension (NE, residues 274–301) of the RdRP is probably necessary to allow the MTase to reach the front of RdRP as we previously suggested [8].

## Naturally fused regions as common regulators of viral RdRPs

Although from the same virus family and utilizing the same *de novo* initiation mechanism, the overall structure is quite different for *Flaviviridae* RdRP molecules from representative virus genera. The 66-kD HCV (the type species of the hepacivirus genus) NS5B comprises the RdRP catalytic core and a 21-residue C-terminal membrane anchor. The 82-kD pestivirus NS5B has a 24-residue C-terminal membrane anchor and a ~90-residue unique N-terminal domain (NTD) specifically modulate the RdRP fidelity through intra-molecular interactions with the RdRP palm [37]. The flavivirus NS5 does not have the C-terminal membrane anchor, but is naturally fused to the capping related MTase at its N-terminus. Compared to the NTD-RdRP regulation in pesitivirus NS5B, the crosstalk between the flavivirus MTase and RdRP seems to be more versatile with respect to conformational diversity and functional relevance, as discussed above.

While the *Flaviviridae* RdRPs exhibit diversities in global structure and regulatory mechanisms involving the naturally fused regions of the RdRP module, this phenomenon appears to be commonly occurring in viral RdRPs [44]. The *Rhabdoviridae* L proteins contain four additional functional regions including an MTase and a polyribonucleotidyl transferase (PRNTase) [45]; the *Bunyavirales* L proteins include two additional regions with analogy to the PA and PB2 subunits of the *Paramyxorviridae* RdRP complex that participate in the cap-snatching process to generate a capped primer for RdRP synthesis [46,47]; the *Alphatetraviridae* RdRPs contain an MTase and a helicase at the N-terminal region [48]; the *Nidovirales* RdRPs, including the *Coronaviridae* nsp12 and *Arteriviridae* nsp9, contain a ~200-400-residue N-terminal region with the nucleotidyltransferase (NiRAN) function [49,50]. Likewise, viral RdRPs may evolve from common ancestors comprising only the catalytic module with relative

independency in carrying out RNA synthesis, similar to the PV 3D$^{pol}$ and HCV NS5B. Co-evolution with versatile host species and the low coding capacity-driven protein function combination may contribute to the structural and functional diversity of current viral RdRPs with respect to regions beyond the RdRP catalytic module. We propose that the functional roles currently offered by these regions, such as the fidelity modulation in pestivirus NS5B and initiation enhancement flavivirus NS5, are likely a result of their co-evolution with the RdRP module through the establishment of specific interactions. Moreover, the RdRP fidelity and initiation enhancement offered by the pesitvirus NS5B NTD and flavivirus NS5 MTase, respectively, is likely not an improvement of a specific function relative to the corresponding RdRP ancestors, but a balance between gaining an extra module and maintaining the levels of key enzymatic properties for the RdRPs.

### Implications for conformation-based protein function

Large-scale protein conformational changes are amazing events in biological systems, albeit associated with very different characteristics. Driven by multiple cycles of phosphoryl transfer reactions and accompanied by dramatic changes in protein-nucleic acid interactions, the N-terminal one third of bacteriophage T7 RNA polymerase undergoes a dramatic rearrangement as it makes an irreversible transition from the promoter-bound initiation state to the promoter-free elongation state in the transcription process [51–53]. By contrast, the observed conformational diversity in flavivirus NS5 has not yet involved continuous events of nucleotide addition, and the energy barrier between different observed states may be low enough to allow a distribution of several states in solution. As both captured by crystallography in multiple virus species, the JEV-mode and DENV3-mode conformations probably represent relatively stable states of apo NS5, thus forming the foundation for further understanding of the NS5 function when it is participating in enzymatic reactions or interactions with essential viral or host factors. The flavivirus NS5 is also a great example that the different interaction modes can be established between regions of a single protein, and are related to different functions of the protein and different processes in the life cycle of the corresponding species. In a broader context, the conformation-based function diversity extends the function capacity beyond the traditional consideration of protein folding, and is therefore an important factor when understanding protein function.

## Materials and methods

### Cloning and protein production

The full-length WT DENV2 NS5 gene within the DNA clone of TSV01 strain (GenBank: AY037116) was cloned into a pET26b vector to yield the pET26b-DENV2-NS5 plasmid. Eleven full-length NS5 constructs with point mutations (Fig 3E) were made by using the QuickChange site-directed mutagenesis method and the WT plasmid as the template. NS5 expressing plasmids were transformed into *Escherichia coli* strain BL21-CodonPlus(DE3)-RIL for expression of NS5 constructs with a hexa-histidine tag at the C-terminus. Cells were grown at 30˚C overnight in the NZCYM medium containing 25 μg/mL kanamycin (KAN25) and 20 μg/mL chloramphenicol (CHL20) until the optical density at 600 nm (OD$_{600}$) was 1.0. The overnight culture was used to inoculate 1 L of NZCYM medium with KAN25 and CHL20 to reach an initial OD$_{600}$ around 0.025. The cells were grown at 37˚C at 220 rpm to an OD$_{600}$ of 1.0 and then cooled to room temperature (r.t.). Isopropyl-β-D-thiogalactopyranoside (IPTG) was added to a final concentration of 0.5 mM, and the cells were grown for an additional 6 h at r. t. or 20 h at 16˚C before harvesting.

## Purification of DENV NS5 and its variants

Cell lysis, subsequent purification and storage procedures were as previously described in the JEV NS5 study [8], except that the centrifugation duration to remove cell debris was 1 h, a 50 mM imidazole wash was applied prior to the elution step of the nickel-affinity chromatography, and 5 mM Tris (pH 7.5) was used as the buffering agent in the gel filtration chromatography. Tris-(2-carboxyethyl)phosphine (TCEP) was added to the pooled fractions to a final concentration of 5 mM. The molar extinction coefficient for the DENV NS5 was calculated based on protein sequence using the ExPASy ProtParam program (http://www.expasy.ch/tools/protparam.html). The yield is typically 2 mg of pure protein per L of bacterial culture.

## Protein crystallization and crystal harvesting

The DENV NS5 crystals were grown by sitting drop vapor diffusion at 10 or 16˚C using 10–12 mg/mL protein sample. Crystals grew to its final dimension in about 3 weeks with a precipitant/well solution containing 1.7% (vol./vol.) dioxane, 0.085 M bicine (pH 8.8), 4.9% (wt./vol.) PEG2000, and 15% (vol./vol.) glycerol for crystal form 1, and 0.2 M $NH_4I$, 8% (vol./vol.) Tacsimate (pH 6.1), and 20% (wt./vol.) PEG3350 for crystal form 2. Crystals were transferred into cryo-stablizer solutions and stored in liquid nitrogen prior to data collection.

## Crystallographic data processing and structure determination

Diffraction datasets were collected at the Shanghai Synchrotron Radiation Facility (SSRF) beamlines BL17U1 (crystal form 1, wavelength 0.9792 Å) and BL18U1 (crystal form 2, wavelengths: 0.9788 Å) at 100 K. Typically, at least 180˚ of data were collected in 0.4–0.5˚ oscillation steps. Reflections were integrated, merged, and scaled using HKL2000 or D*Trek [54,55]. The initial structure solution for crystal form 1 was obtained using the molecular replacement program PHASER [56] and separated MTase and RdRP ensembles derived from JEV and DENV3 NS5 structures (PDB entries 4K6M and 4V0Q) [8,20]. The final model of crystal form 1 was split into three ensembles (MTase, RdRP thumb with index tip, and the rest of RdRP) in the molecular replacement trial to obtain the initial structure solution for crystal form 2. Manual model building and structure refinement were done using Coot and PHENIX, respectively [57,58]. The 3,500 K composite simulated-annealing omit $2F_o$-$F_c$ electron density maps were generated using CNS [59]. All NS5 superimpositions were done using the maximum likelihood based structure superpositioning program THESEUS [26]. Relative domain motions were analyzed by DynDom [60]. The occlusion of the solvent accessible area by the MTase-RdRP interactions was analyzed by program SurfRace with a probe radius of 1.4 Å and the 10-residue linker excluded in the calculation [61]. The multiple sequence alignment of NS5 was carried out using 47 available complete NS5 sequences among the flavivirus species documented by the International Committee on Taxonomy of Viruses (ICTV) (http://www.ictvonline.org), and the alignment was then used to generate the sequence logos (http://weblogo.berkeley.edu) and the conservation score projected structural representations. The projection of the conservation score onto structural model was done by the ConSurf server [62].

## Cells and antibodies

BHK-21 cells (American Type Culture Collection (ATCC), CCL-10) was propagated in Dulbecco's modified Eagle's medium (DMEM) supplemented with 10% fetal bovine serum (FBS), 100 units/mL of penicillin and 100 μg/mL of streptomycin in 5% $CO_2$ at 37˚C. Monoclonal antibody (4G2) against envelope protein of flavivirus was used to detect viral protein expression of both JEV and DENV2. FITC-conjugated goat anti-mouse IgG was used as secondary antibody.

## Construction of mutant genome-length cDNA of JEV and DENV2

The infectious clones of pACYC-JEV-SA14 [63] and pACYC-DENV2-TSV [33] were used as the backbone to construct recombinant JEV and DENV2 with different NS5 mutations, respectively. All mutations were engineered by fusion PCR. The JEV NS5 mutations were engineered at *BamH*I and *Xba*I restriction sites of pACYC-JEV-SA14. All DENV2 NS5 mutations were inserted into to pACYC-DENV2-TSV by restriction digestion with *Nru*I and *Cla*I. All constructs were verified by DNA sequencing before they were used in the subsequent experiments.

## Recombinant JEV and DENV2 study with different NS5 mutations

The infectious clone of JEV and DENV2 with corresponding NS5 mutations were linearized with *Xho*I and *Cla*I, respectively, and then subjected to *in vitro* transcription using a T7 *in vitro* transcription kit (Thermo Fisher Scientific). Approximately 1 μg of transcribed recombinant genomic RNAs were transfected into BHK-21 cells with reagent DMRIE-C (Invitrogen). Then the cell slides were fixed in cold (-20˚C) 5% (vol. to vol.) acetone in methanol at r. t. for 10 min. After washing three times with phosphate buffer saline (PBS) (pH 7.4), the fixed cells were subjected to IFA with 4G2 monoclonal antibody to examine viral envelope expression of both JEV and DENV2. At the same time, the supernatants of RNA-transfected BHK-21 cells were harvested as viral stocks for plaque assay to quantify viral titers and examine plaque morphologies. Briefly, confluent BHK-21 cells ($1\times10^5$ cells per well, plated 1 day in advance) in 24-well plates were infected with serially 10-fold diluted viral supernatants and incubated at 37˚C with 5% $CO_2$ for 1 h before the layer of medium containing 1% methylcellulose was added. After 4 days of incubation at 37˚C with 5% $CO_2$, the cells were fixed in 3.7% formaldehyde and then stained with 1% crystal violet. The viral titer was calculated as plaque formatting unit (PFU) per mL.

## RNA preparation

The 30-mer template RNA (T30, Fig 5A) used for *de novo* polymerase assays was chemically synthesized (Integrated DNA Technologies or Dharmacon) and purified by 12% (wt./vol.) polyacrylamide/7 M urea gel electrophoresis. The target RNA was excised from the gel, electro-eluted by using an Elu-Trap device (GE Healthcare), ethanol precipitated, dissolved in an RNA annealing buffer (RAB: 50 mM NaCl, 5 mM Tris (pH 7.5), 5 mM $MgCl_2$), and stored at -80˚C after a self-annealing process (a 3-min incubation at 95˚C followed by snap-cooling to minimize inter-molecular annealing). A 5′-phosphorylated dinucleotide primer pGG (P2) (Jena Biosciecnes) was mixed with T30 at 5:1 or 20:1 molar ratio to make the T30/P2 construct.

## *In vitro* polymerase assays

All *in vitro* polymerase assays were based on the dinucleotide (P2)-driven reactions. The standard reaction condition was derived from the JEV NS5 work with the 4 μM of the T30 RNA and 6 μM of NS5 and with a couple of adjustments [12]. Firstly, a manganese-free reaction condition (50 mM Tris (pH 7.5), 20 mM NaCl, 5 mM $MgCl_2$, 5 mM DTT) (hereinafter referred as reaction buffer) was established. Secondly, a 5′-phosphorylated dinucleotide (pGG) and the T30 RNA was mixed at a difference ratio. In the assay to characterize the conversion of P2 to the 3-mer product (P3), ATP was supplied as the only NTP substrate and a high P2:T30 ratio (20:1) was used to achieve multiple turnovers within a reasonable duration. For the P9-containing EC (EC9) formation assay to achieve the P2-to-P9 conversion, ATP and UTP

were supplied and the P2:T30 ratio was 5:1. For the P9-to-P10 single nucleotide elongation assay, reactions were first carried out as described in the EC9 formation assay. The reaction mixtures were centrifuged at 16,000 g for 5 min, and the pellet was washed twice by the reaction buffer, and was then resuspended in a modified reaction buffer with NaCl concentration lifted to 200 mM. CTP was supplemented to the resuspended mixture to allow the single-nucleotide elongation at 30˚C, and the reaction was quenched immediately ("0" min) or at a certain time point after the addition of CTP. The concentrations of NaCl and CTP in the final reaction mixture are 200 mM and 300 μM, respectively, unless otherwise indicated. For all assays, the procedures for denaturing polyacrylamide gel electrophoresis (PAGE), gel staining, and quantitative analyses were performed as previously described [12]. The Stains-All (Sigma-Aldrich)-based staining method is reasonably accurate when quantitating RNA bands with the same length, and in the majority of our experiments we tried to keep the band intensity within the linear range estimated in the previous study by adjusting the range of reaction time points [12].

For the P2-to-P3 conversion assay, 50, 100, 200, 500, and 1000 μM ATP concentrations were used for WT and M-67A/68A, and 200, 400, 600, 1000, and 1500 μM were used for R3. To account for gel-to-gel intensity variations, samples from the same reaction mixture (indicated by the same icon above corresponding lanes in Fig 7A–7C) were loaded on different gels to normalize the intensities (e.g. lanes 4/17, 14/27, 24/37 in Fig 7A). The normalized intensity was then used to calculate the relative reaction rates (Fig 7D–7F, left), which in turn were fitted to the Michaelis-Menten equation (Fig 7D–7F, right). To estimate the relative $k_{cat}$ and specificity constant values of all three constructs, The Michaelis-Menten curves of all three constructs were normalized based on the measurement of the relative reaction rates of each WT-mutant pair in the same gel (Fig 7G).

For the stability assessment of EC9, the P2-to-P9 conversion was first carried out as described above. The reaction mixture was subsequently centrifuged at 16,000 g for 5 min, and the pellet was washed twice by the reaction buffer and was then resuspended in a modified reaction buffer with NaCl concentration lifted to 500 mM. The resuspension was incubated in 37˚C for various duration (0–78 h), and then CTP was added to reach a final concentration of 300 μM to trigger the P9-to-P10 conversion by EC9 that survived the incubation. The fraction of 10-mer intensity values were fitted to a single-exponential decay model to estimate the inactivation rates of all three constructs.

## Supporting information

**S1 Movie. A modelled NS5 conformational transition from the JEV-mode to the DENV3-mode.** The movie starts with the JEV NS5 structure (PDB entry 4K6M), switches to the first form of DENV2 structure (PDB entry 6KR2), then the second form of DENV2 structure (PDB 6KR3), and finally the DENV3 structure (PDB entry 4V0Q). Three rotation axes related to the MTase movement are indicated by black lines, and one axis related to the RdRP thumb movement is indicated by a grey line. Note that the non-superimposable regions are not included in the structural models.
(MP4)

## Acknowledgments

We thank Dr. Pei-Yong Shi for providing the cloning material for the DENV2 NS5 gene, Dr. Xiao-Dan Li for construction of the DENV2 NS5 M3 mutant plasmid, Dr. Bo Shu for X-ray diffraction data collection, Liu Deng and Yancheng Zhan for laboratory assistance, Dr.

Yunhuang Yang for helpful discussions, synchrotron SSRF (beamlines BL17U and BL18U1 Shanghai, China) for access to beamlines, and The Core Facility and Technical Support, Wuhan Institute of Virology, for access to instruments.

## Author Contributions

**Conceptualization:** Jiqin Wu, Bo Zhang, Peng Gong.

**Data curation:** Jiqin Wu, Bo Zhang, Peng Gong.

**Formal analysis:** Jiqin Wu, Han-Qing Ye, Qiu-Yan Zhang, Guoliang Lu, Bo Zhang, Peng Gong.

**Funding acquisition:** Han-Qing Ye, Peng Gong.

**Investigation:** Jiqin Wu, Han-Qing Ye, Qiu-Yan Zhang, Guoliang Lu.

**Project administration:** Peng Gong.

**Resources:** Bo Zhang, Peng Gong.

**Supervision:** Bo Zhang, Peng Gong.

**Visualization:** Jiqin Wu, Qiu-Yan Zhang, Bo Zhang, Peng Gong.

**Writing – original draft:** Jiqin Wu, Guoliang Lu, Bo Zhang, Peng Gong.

**Writing – review & editing:** Jiqin Wu, Han-Qing Ye, Qiu-Yan Zhang, Bo Zhang, Peng Gong.

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
