## [Decision Letter · Decision Letter 0]

25 Feb 2020

Dear Dr. Gong,

Thank you very much for submitting your manuscript "A conformation-based intra-molecular initiation factor identified in the flavivirus RNA-dependent RNA polymerase" for consideration at PLOS Pathogens. As with all papers reviewed by the journal, your manuscript was reviewed by members of the editorial board and by two independent reviewers. As you will see, the reviewers provided contradictory reports. After careful consideration, we have editorially sided with reviewer 2.   Flavivirus NS5 is a multifunctional protein, composed of an N-terminal methyl-transferase (MTase) domain and a larger RNA-dependent RNA polymerase (RdRp) C-terminal domains. Multiple structures have been reported previously, some identifying a functional dimeric complex. As a polymerase, the NS5 protein has to adopt several conformation, which are different for initiation of RNA synthesis, for capping the newly synthesized RNA, or for elongation of the latter. In your study, you obtained crystals of DENV2 NS5 adopting two different conformations : one that resembles a structure reported for NS5 from the Japanese encephalitis flavivirus (JEV), in which the MTase domain adopts a “closed conformation” with respect to the RdRp domain, and another in which it adopts an "open conformation", similar to the one seen before in a structure of NS5 from the dengue virus serotype 3 (DENV3). As the two MTase/RdRp interfaces are different,  site directed mutagenesis allowed the selective targeting of the two interfaces. The use of these mutants for functional studies in an assay that allows to distinguish RNA initiation from RNA elongation showed that mutations at the interface seen in the open form interfere with the initiation process but do not affect the elongation process. Although there was no effect of the mutants at the interface observed in the closed form, it was considered editorially that it  is the first time that a given conformation of NS5 can be clearly attributed to a specific step in the replication process, and even though no MTase studies were performed, as pointed out by reviewer 1, that this is still a meaningful study.

[1] A letter containing a detailed list of your responses to all review comments, in particular to those raised by reviewer 2, but also, whenever possible, to those raised by reviewer 1, together with a description of the changes you have made in the manuscript. 

Sincerely,

Félix A. Rey

Associate Editor

PLOS Pathogens

Michael Diamond

Section Editor

PLOS Pathogens

Kasturi Haldar

Editor-in-Chief

PLOS Pathogens

orcid.org/0000-0001-5065-158X

Michael Malim

Editor-in-Chief

PLOS Pathogens

orcid.org/0000-0002-7699-2064

The paper describes the X-ray structure of the NS5 protein of Dengue virus serotype 2. Flavivirus NS5 is a multifunctional protein, composed of an N-terminal methyl-transferase (MTase) domain and a larger RNA-dependent RNA polymerase (RdRp) C-terminal domains. Multiple structures have been reported previously, some identifying a functional dimeric complex. As a polymerase, this protein has to adopt several conformation, which are different for initiation of RNA synthesis, for capping the newly synthesized RNA, and for elongation of the later. Here, the authors have identified two different conformations of NS5 in their crystals: one that resembles a structure reported for NS5 from the Japanese encephalitis flavivirus (JEV), in which the MTase domain adopts a “closed conformation” with respect to the RdRp domain, and another in which it adopts an open conformation, similar to the one seen before in a structure of NS5 from the dengue virus serotype 3 (DENV3). What I found novel is that, as the two MTase/RdRp are different, they used directed mutagenesis to alter the interfaces in the closed or in the open conformation. They used these mutants for functional studies in an assay that allowed them to distinguish RNA initiation from RNA elongation. They found that mutations in the interface seen in the open form interfere with the initiation process but do not alter the elongation process. Although they didn’t found a phenotype for the mutants at the interface observed in the closed form, this is the first time that a given conformation of NS5 can be assigned to a particular step in the replication process, and even though no MTase studies were performed, as pointed out by reviewer 1, I found that the authors have still presented a meaningful study.

Reviewer Comments (if any, and for reference):

Reviewer's Responses to Questions

**Part I - Summary**

Reviewer #1: In this manuscript, Wu et al. report the crystal structures Dengue virus type 2 (DENV2) NS5 and show that it exists in two distinct conformations representing both of the previously reported structures of Japanese encephalitis virus (JEV) and DENV3 NS5. These are defined as JEV-mode and DENV3-mode due to closeness of these DENV2 structures to the JEV and DENV3 NS5 structures. Moreover, the authors present additional data by mutational analysis to distinguish the functional differences between the two conformations of DENV2 NS5. They performed in vitro polymerase assays with wild-type (WT) and site-specific NS5 mutants to distinguish the initiation and elongation steps of RNA synthesis as well as cell-based replication using immunofluorescence assays. The authors conclude that the methyltransferase (MTase) of NS5 serves as a unique initiation factor only through NS5 conformation in the JEV-mode. But no MTase assays were performed. Overall, the crystallographic work is well-done. However, their biochemical and virological assays raise several questions. The functional roles of these two conformational states of DENV2 NS5 in viral replication remains unclear. Moreover, another group reported recently that DENV2 NS5 exists in two distinct conformational states (El Sahili, Soh, Schiltz, Gharbi-Ayachi, She, Shi, Lim, Lescar, 2020. J. Virol. 94: e01294-19). These authors in El Sahili et al. provide convincing evidence that it is the inter-domain linker between the MTase and POL domains the functional determinant of the two conformational states of DENV2 NS5.

Reviewer #2: The flaviviral NS5 proteins contain a methyltransferase (MTase) domain followed by an RNA-dependent RNA polymerase (RdRP) domain, both of which have essential functions during viral replication. Structures of the individual domains have been known for over a decade and biochemical approaches have been applied to studying the seemingly independent activities of these domains; this has worked moderately well for the MTase, but not so well for the RdRP due to the presence of a priming loop on the thumb domain that limits RNA binding and has precluded detailed biochemical studies of elongation kinetics the solving of an active elongation complex structure. Over the past few years that have several structure of full length NS5 solved from multiple flaviviruses (JEV, ZIKV, DENV3), and interestingly these have shown different interactions between the MTase and RdRP domains. However, it has not yet been clear if the observed inter-domain interactions are biologically relevant or due to crystal packing effects.

The work presented in this manuscript provides a thorough study of MTase-RdRP interactions using structural biology, biochemistry, and virus replication data to address functional roles of the flaviviral methyltransferase (MTase) during the initiation of replication by the RdRP domain. Two new structures of DENV2 NS5 are solved that capture domain orientations which are intermediate between those solved previously for DENV3 and JEV. Structure-based mutagenesis is then used to show that one specific mode of MTase-RdRP interaction (the JEV mode) is important for both the growth of infectious virus and for in vitro initiation by the RdRP domain. These new results provide key insights into understanding the molecular mechanisms whereby flaviviral NS5 initiates replication.

Using initiation of a pGG dinucleotide to form pGGA or to extent to a longer 9mer when UTP is also added, the authors use mutations of the show that formation of both products is diminished upon mutation of the JEV-mode interface but not the DENV3 interface, indicating that the JEV-mode is the active form. Furthermore, the researchers demonstrate that mutating the MTase interaction only affects RdRP initiation to form short products (pGG to pGGA), and does not affect on the elongation of a longer 9mer product to a 10mer. This latter elongation step is demonstrated to be very rapidly done by a complex that is stable in high salt, showing that the authors have assembled authentic stalled elongation complexes form a flaviviral polymerase. This is a major breakthrough that sets the stage for future studies of both RdRP biochemistry and elongation complex structure.

The core experiments and conclusions are solid and well presented in the manuscript and there are no major concerns with the interpretation of the results. As usual from Gong lab, the illustrations are excellent and the polymerase biochemistry is well done with rigorous interpretations, all the more notable in this work because of the extensive steps taken to normalize Stains-All data across multiple gels. However, there are a few points where additional clarification or grammar corrections would be appreciated, as outlined below.

Line 146-149: It may be worth discussing that the ring finger and part of index finger are not resolved in many of the early flaviviral RdRP domain structures (2hcn, 2hfz, 2j7u, 2j7w… see the flaviviral structures at DOI: 10.5281/zenodo.3361874). Might this be because the MTase domain is needed to stabilize the fingers folding, which could have big effects on RNA binding and initiation?

The section from lines 162-167 describing SAH binding site is weak and the evidence for an “allosteric” site sounds somewhat circumstantial. Is there any experimental evidence for SAH binding being important, or is it perhaps a crystallization artifact? The section should either be strengthened with experimental data or references, or the idea of “allosteric” should be removed and the observed SAH density instead be described as an interesting structural observation that merits further biochemical validation. Also, was SAH added to the crystallization conditions, or did this SAH co-purify with the protein much as the SAH bound to the MTase domain does? Is there full occupancy of this “allosteric” site and of the native MTase site? Is there any (weak) anomalous density from the sulfur that would definitively identify the SAH?

Please explicitly state the actual concentration of polymerase, RNA template and pGG primer used in the biochemical experiments. These are not stated anywhere in the manuscript, where only mole ratios are listed.

**Part II – Major Issues: Key Experiments Required for Acceptance**

Reviewer #1: In this manuscript, Wu et al. report the crystal structures Dengue virus type 2 (DENV2) NS5 and show that it exists in two distinct conformations representing both of the previously reported structures of Japanese encephalitis virus (JEV) and DENV3 NS5. These are defined as JEV-mode and DENV3-mode due to closeness of these DENV2 structures to the JEV and DENV3 NS5 structures. Moreover, the authors present additional data by mutational analysis to distinguish the functional differences between the two conformations of DENV2 NS5. They performed in vitro polymerase assays with wild-type (WT) and site-specific NS5 mutants to distinguish the initiation and elongation steps of RNA synthesis as well as cell-based replication using immunofluorescence assays. The authors conclude that the methyltransferase (MTase) of NS5 serves as a unique initiation factor only through NS5 conformation in the JEV-mode. But no MTase assays were performed. Overall, the crystallographic work is well-done. However, their biochemical and virological assays raise several questions. The functional roles of these two conformational states of DENV2 NS5 in viral replication remains unclear. Moreover, another group reported recently that DENV2 NS5 exists in two distinct conformational states (El Sahili, Soh, Schiltz, Gharbi-Ayachi, She, Shi, Lim, Lescar, 2020. J. Virol. 94: e01294-19). These authors in El Sahili et al. provide convincing evidence that it is the inter-domain linker between the MTase and POL domains the functional determinant of the two conformational states of DENV2 NS5.

Specific comments

1. Line 71, p. 4: This study focuses only on the polymerase activity of NS5 although the interdomain interaction between MTase and POL domains could impact on both enzyme activities. Why the authors did not perform any MTase assays to study the impact of the two conformations of DENV2 NS5?

2. Lines 99-101, p. 5: “This statement is not correct. Once initiated de novo, RdRp can continue RNA synthesis until the template is copied. This has been demonstrated for dengue virus type 2 RdRp in many studies using subgenomic RNA templates containing 5’- and 3’-terminal regions and conserved cis-active elements that have been shown to be important for both initiation and elongation steps (see refs. Ackermann and Padmanabhan, 2001; Niyomrattanakit et al. 2010). The statement on line 101 is not accurate because in one study, processivity of RdRp was demonstrated by including heparin after initiation complex is formed to block re-initiation events (Nomaguchi et al. 2003). The statement on line 102 needs to modified to indicate that both Mg++ and Mn++ were used (in refs. 18, 24 and 25 cited in this study).

3. The effects of mutations on MTase activity were not studied whereas these were examined in ref. 18 cited in this study.

4. Lines 204, 208, 264: There is no Fig. 3E included in the manuscript.

5. Lines 235-239 and lines 239-242: Please cite the ref(s) here; 12 or 33?.

6. One concern with this study is the use of artificial GG dinucleotide primer and template for RdRp initiation and elongation assays. The authors’ definition of de novo initiation in mosquito-borne RdRp assays is unconventional and may lead to wrong conclusions regarding the mutational effects in RdRp activities. For example, the catalytic rate constant could not be accurately determined under their conditions (lines 252-257).

Reviewer #2: None

**Part III – Minor Issues: Editorial and Data Presentation Modifications**

Reviewer #1: Minor Comments:

1. Line 144: Which DENV3 structures the authors are referring to here? It would be helpful if the authors label the motif G in Fig. 1.

2. Lines 160-162: The SAH binding site has been observed and labeled in Klema et al. (ref. 21).

3. Define “pt” ( line 229) andNS5B NTD (line 403).

Reviewer #2: Line 40: calling differences in prior structures “drastically different global folds” is perhaps an overstatement since the effects are really at the level of relative domain orientations and not protein folding.

Line 51: remove “about” as it redundant with giving a range of 10-11 kb for genome length.

Line 71: change are to is

Line 72: make evidences singular

Line 76: typo in hereafter

Line 86: in discussing interface areas, please explicitly state if the values presented reflect total buried solvent accessible surface from both sides of the interface, or from only one side of the interface. Also on lines 138, 157 and others.

Line 115: Consider deleting “apparently” as the data for this are convincing.

Line 120: change in to to.

Line 127: arrange should be in past tense (arranged).

Line 301: The same high-salt challenge studies are also used in studies of picornaviral polymerases, where they led to the crystallization of elongation complexes.

Line 312-316: What really needs to the optimized here is the amount of material loaded on the gel, not necessarily what was used in the reactions, which could be done at higher concentrations and then diluted prior to gel loading (a minor conceptual point, no need to edit the text).

Line 316: Change 3-fold to “3-fold higher”

Line 375: I do not agree with the statement that the global structural organization is quite different from flaviviral polymerases versus others. Structure comparisons have shown the core polymerase fold and structure to be very similar across all viral polymerases, and certainly among positive strand RNA viruses, with few insertions or topology changes within the core fold. There are added domains, such as the N-terminal MTase and NiRAN domains or C-terminal membrane anchors, but these are additions to the protein and not a reorganization of the protein fold as implied by the manuscript as written.

Line 394: I do believe there is convincing evidence that the N-terminal region of nsp9 is a nucleotidylating NiRAN domain.

Line 518: Include actual concentrations of macromolecules used in the experiments.

Figure 3: Are the stereo panels in A and B flipped right to left? Stereo viewing of these images does not seem to work correctly.

Figure 5: In panel D, why is the 9mer STD band higher than the 9mer product band from the polymerase? Effect of the 5’ phosphate from pGG primer, perhaps?

Line 816: Consider changing to “…in the multiple turnover P3 formation and single turnover P9 formation assays” so as to explain the vast difference in band intensities in the figure legend itself (in addition to the comments already in the text of the manuscript).

Line 833: Technically speaking, the experiment in really testing inactivation of the elongation complex and not the dissociation rate of the RNA, although that is probably a valid interpretation in this case.

Figure 7: In some cases, linear curve fits to determine initial rates are being applied to data that have clear curvature (panel E-200uM, panel F-50 & 100 uM, panel G-R3). This is a minor point.

PLOS authors have the option to publish the peer review history of their article (what does this mean?). If published, this will include your full peer review and any attached files.

Reviewer #1: No

Reviewer #2: No
---

## [Editor Report · Decision Letter 1]

18 Mar 2020

Dear Dr. Gong,

We are pleased to inform you that your manuscript 'A conformation-based intra-molecular initiation factor identified in the flavivirus RNA-dependent RNA polymerase' has been provisionally accepted for publication in PLOS Pathogens.

Best regards,

Félix A. Rey

Associate Editor

PLOS Pathogens

Michael Diamond

Section Editor

PLOS Pathogens

Kasturi Haldar

Editor-in-Chief

PLOS Pathogens

orcid.org/0000-0001-5065-158X

Michael Malim

Editor-in-Chief

PLOS Pathogens

orcid.org/0000-0002-7699-2064
---

## [Editor Report · Acceptance letter]

24 Apr 2020

Dear Dr. Gong,

We are delighted to inform you that your manuscript, "A conformation-based intra-molecular initiation factor identified in the flavivirus RNA-dependent RNA polymerase," has been formally accepted for publication in PLOS Pathogens.

Best regards,

Kasturi Haldar

Editor-in-Chief

PLOS Pathogens

orcid.org/0000-0001-5065-158X

Michael Malim

Editor-in-Chief

PLOS Pathogens

orcid.org/0000-0002-7699-2064